# Over the Counter Pain Medications Used by Adults: A Need for Pharmacist Intervention

**DOI:** 10.3390/ijerph20054505

**Published:** 2023-03-03

**Authors:** Katarzyna Karłowicz-Bodalska, Natalia Sauer, Laura Jonderko, Anna Wiela-Hojeńska

**Affiliations:** 1Department of Drugs Form Technology, Faculty of Pharmacy, Wroclaw Medical University, 50-556 Wroclaw, Poland; 2Department of Clinical Pharmacology, Faculty of Pharmacy, Wroclaw Medical University, 50-556 Wroclaw, Poland

**Keywords:** pharmaceutical care, non-opioid analgesics, over-the-counter medications, geriatric, adverse drug reactions

## Abstract

Background: The safety of pharmacotherapy for geriatric patients is an essential aspect of the demographic perspective in view of the increasing size of this population. Non-opioid analgesics (NOAs) are among the most popular and often overused over-the-counter medications (OTC). The reasons for drug abuse are common in the geriatric population: musculoskeletal disorders, colds, inflammation and pain of various origins. The popularity of self-medication and the ability to easily access OTC drugs outside the pharmacy creates the danger of their misuse and the incidence of adverse drug reactions (ADRs). The survey included 142 respondents aged 50–90 years. The relationship between the prevalence of ADRs and the NOAs used, age, presence of chronic diseases, and place of purchasing and obtaining information about the mentioned drugs were evaluated. The results of the observations were statistically analyzed using Statistica 13.3. The most commonly used NOAs among the elderly included paracetamol, acetylsalicylic acid (ASA) and ibuprofen. Patients consumed the medications for intractable headaches, toothaches, fevers, colds and joint disorders. Respondents indicated the pharmacy as the main location for purchasing medications, and the physician as the source of information for selecting the therapy. ADRs were reported most frequently to the physician, and less frequently to the pharmacist and nurse. More than one-third of respondents indicated that the physician during the consultation did not take a medical history and did not ask about concomitant diseases. It is necessary to extend pharmaceutical care to geriatric patients that includes advice on adverse drug reactions, especially drug interactions. Due to the popularity of self-medication, and the availability of NOAs, long-term measures should be taken to increase the role of pharmacists in providing effective, safe health care to seniors. We are targeting pharmacists with this survey to draw attention to the problem of the prevalence of selling NOAs to geriatric patients. Pharmacists should educate seniors about the possibility of ADRs and approach patients with polypragmasy and polypharmacy with caution. Pharmaceutical care is an essential aspect in the treatment of geriatric patients, which can contribute to better results in their existing treatment and increase the safety of medication intake. Therefore, it is important to improve the development of pharmaceutical care in Poland in order to enhance patient outcomes.

## 1. Introduction

Recent years have seen a steady increase in the elderly population. It is estimated that, by 2050, the number of people over 60 years of age worldwide will reach two billion. In 2020, Poland’s population was 38.386 million, including 7.1 million (18.6%) of the population over 65. According to demographic projections by the Central Statistical Office, by 2050, this percentage may even increase to 32% [1,2,3].

In Poland, the elderly regularly take an average of seven medications, including more than five prescription drugs. These data do not differ from observations coming from other countries. Thürmann et al. demonstrated that about 42% of people aged 65 and older take five or more medications regularly [4]. Polypharmacotherapy is a consequence of the multimorbidity that characterizes this age group in up to 62% of people. As the number of drugs used increases, so does the risk of side effects, including interactions, especially at the metabolic stage [5,6]. Specifically, in a patient taking five to nine medications, the probability of ADR is about 50%, whereas the risk increases to 100% when the patient takes 10 or more medications [7,8]. The geriatric population taking multiple medications is additionally more susceptible to TV commercials [9,10]. Consequently, polytherapy often escalates into polypragmasy, meaning the medically unjustified, irrational consumption of more drugs. As a result, it is observed that patients take multiple analgesics, resulting in the synergism of their effects. The intensified effect of drugs leads to drug toxicity and the occurrence of adverse drug reactions.

Among the most commonly used OTC drugs in Poland are non-opioid analgesics, used in the treatment of musculoskeletal disorders, febrile conditions, inflammation and the common cold [11,12,13,14,15] As many as 96% of patients over the age of 65 take them in general practice [16]. It has also been shown that about 7.3% of people over the age of 60 fill at least one prescription for a non-opioid analgesic per year [17,18,19].

Due to the widespread availability of these medications and the risks associated with their incorrect use, as well as the difficulty of contacting a physician, it is necessary to effectively implement and develop pharmaceutical care, especially for elderly patients. The purpose of this study was to analyze the safety of non-opioid analgesics used by the elderly and the importance of the pharmacist in optimizing pain management in geriatric patients. In this paper, we examine the most common adverse drug reactions associated with the use of NOAs and identify factors that intensify drug toxicity in elderly patients. We highlight the level of knowledge of older patients regarding pharmaceutical care and their willingness to access it. 

## 2. Material and Methods

### 2.1. Data Source

The observations were carried out using an anonymous self-formulated questionnaire, which served the research project, entitled “Evaluation of the safety of NOA use in the geriatric population”, approved by the Bioethics Committee operating at Wroclaw Medical University (Appendix A). In the survey, we used a form consisting of 88 single- and multiple-choice questions to collect data from patients. The questionnaire included questions about the reasons for the use of pain medications, comorbidities, the use of dietary supplements, the source of purchasing and obtaining information about medications, and knowledge of reporting adverse drug reactions. Initially, we collected data using an online questionnaire, but due to the low online activity of people over 65, data were also collected at pharmacies.

### 2.2. Patients Group

The study included 142 people aged 50–90 years. For the study, participants with preserved cognitive ability taking medication on their own were eligible. Among the surveyed population, 62% (n = 88) were female and 38% (n = 54) male. The largest group 50.7% (n = 72) was between 50 and 60 years of age, 38.7% (n = 55) between 61 and 75 years of age, and 10.6% (n = 15) between 76 and 90 years of age (Figure 1). 

Diabetes was statistically significantly more common in men 37% (n = 53) than in women 19.3% (n = 27). Women were significantly more likely to report complaints of depression and neurosis 26.1% (n = 37) compared to 7.4% (n = 10) of men. Detailed data are shown in Table 1.

Among the study group, cardiovascular disease, hypertension, diabetes, and renal failure were significantly more common among the subjects over 75 years of age. Obesity was significantly more common in the 61–75 age range. These relations are shown in Table 2.

### 2.3. Statistical Analysis

The results of the observations were subjected to statistical analysis using Statistica 13.3. The probability of correlation between variables was supported using count tables and multivariate tables. Statistical significance was assessed using Pearson’s χ^2^ test of concordance, taking α = 0.05 as the level of significance. In the case of expected counts of less than 5, the chi-square test with Yates correction was applied additionally. 

## 3. Results

Observations showed that NOAs were used most often for headache and toothache (89.4%) (n = 127), fever and cold (88%) (n = 125) and joint diseases (27.5%) (n = 39) and less often as a preventive measure against myocardial infarction (9.2%) (n = 13) and to treat gout (7%) (n = 10). In the population suffering from joint disease, NOAs were more frequently used by women than men (χ^2^ = 5.100, *p* = 0.024). The most commonly used were paracetamol, ibuprofen, acetylsalicylic acid (ASA), ketoprofen and diclofenac. Most often—several times a week—respondents took ibuprofen, and less often, nimesulide, acetylsalicylic acid, paracetamol, ketoprofen and naproxen. Less than once a week, respondents used paracetamol, ASA, ketoprofen, diclofenac, naproxen and metamizole. The prevalence of non-opioid analgesic use among the elderly is shown in Figure 2. 

ADRs were observed by 28% (n = 40) of respondents, with the majority reporting to the physician 75% (n = 30), 20% (n = 8) to the pharmacist, and only 5% (n = 2) to the nurse. With topical skin application, ADRs included local skin reactions, which were significantly more common in respondents aged 61–75 years 5.6% (n = 8) (χ^2^ = 6.644 *p* = 0.036). NOA users in the 50–60 age range (19.4%, n = 14) were significantly more likely to report gastrointestinal disorders, while respondents aged 61–75 were significantly more likely to complain about allergic reactions (14.8%, n = 8.)

Dietary supplements were used by 41.5% (n = 59) of respondents, mainly omega-3 acids, and preparations containing St. John’s wort and ginkgo biloba. In patients taking a combination of dietary supplements and NOA, the incidence of ADRs increased significantly. The population taking dietary supplements in combination with ASA or paracetamol had a higher incidence of gastrointestinal bleeding (χ^2^ = 4.738 *p* = 0.029), (χ^2^ = 11.672 *p* = 0.001), while the combination with ketoprofen had a higher incidence of gastrointestinal disorders (χ^2^ = 5.073 *p* = 0.024).

Respondents who did not check the expiration date of the drug before taking it were significantly more likely to observe gastrointestinal bleeding after using ASA (31.3%) (n = 10) (χ^2^ = 8.084 *p* = 0.004) or ibuprofen (15.6%) (n = 5) (χ^2^ =9.804 *p* = 0.002).

The vast majority of respondents (95.1%) (n = 135) bought their medications from a pharmacy, and most (67%) (n = 95) got their information from a doctor, less often from a pharmacist (44%) (n = 62) and (22.5%) (n = 32) from the media.

Taking medications was significantly more often forgotten by men (72.2%) (n = 102.5), versus 53.4% (n = 75.8) of women (χ^2^ = 4.959 *p* = 0.026). Women were more likely to skip a dose they did not take, while men took it when they remembered it. Women used drugs after eating, while men were significantly more likely to use them on an empty stomach (χ^2^ = 8.994 *p* = 0.011). Respondents sipping medication with a beverage other than water (17.4% n = 4), were more likely to report gastrointestinal bleeding (χ^2^ = 5.091 *p* = 0.024). 

A self-reported survey shows that many patients did not receive a proper medical interview and advice. As many as 33.9% (n = 48) of respondents indicated that the doctor during the consultation did not take a history and did not ask about concomitant diseases. It is worrying that, despite the concerns, a significant proportion of patients did not ask for medical advice from a pharmacist.

### 3.1. Analysis of the Safety of Paracetamol Therapy

Paracetamol was used by 76.7% (n = 109) of respondents, including several times a week by 14.1% (n = 20) and less than once a week by 62.7% (n = 89). Gastrointestinal bleeding (36.84%, n = 7) and gastrointestinal upset (57.9%, n = 11) were significantly more frequent when the drug was used several times a week (χ^2^ = 33.576 *p* = 0.001). Of the respondents with peptic ulcer disease, 21.8% (n = 7) reported gastrointestinal bleeding (χ^2^ = 22.718 *p* = 0.0004). Attention to polytherapy is also significant, as respondents taking concurrent paracetamol and two or more NOAs were more likely to observe gastrointestinal bleeding (χ^2^ = 4.552 *p* = 0.033).

### 3.2. Analysis of the Safety of Ibuprofen Therapy

Ibuprofen was used by 69.0% (n = 98) of respondents, taken several times a week by 12% (n = 17) and less than once a week by 57% (n = 81). Regular smokers were more likely to report gastrointestinal bleeding (7.1%, n = 16), (χ^2^ = 22.846 *p* = 0.00004) and so too were alcohol consumers (7.1%, n = 16), (χ^2^ = 5.500 *p* = 0.019). Respondents taking the drug before meals (12%, n = 3) were significantly more likely to observe gastrointestinal bleeding (χ^2^ = 6.719 *p* = 0.035). Gastrointestinal bleeding was significantly more common in those who did not check the expiration date of the drug before taking it 15.6% (n = 5, χ^2^ = 9.804 *p* = 0.002). Gastrointestinal disorders, gastric ulcers, duodenal ulcers and allergic reactions were most frequently reported. 

### 3.3. Analysis of the Safety of Acetylsalicylic Acid (ASA) Therapy

ASA was used by 60% (n = 86) of respondents, most often in the 50–60 age group, with 15.5% (n = 22) using it several times a week and 45.1% (n = 64) using it less than once a week. Alcohol consumers were significantly more likely to observe gastrointestinal bleeding, dizziness and tinnitus (χ^2^ = 5.699 *p* = 0.017). Those taking ASA on an empty stomach were also significantly more likely to report dizziness and tinnitus (χ^2^ = 17.262 *p* = 0.00004). Gastrointestinal bleeding after ASA use was significantly more common when the drug was taken several times a week (χ^2^ = 7.755 *p* = 0.005). In those with peptic ulcer disease, ASA therapy correlated with more frequent ADRs, reported gastrointestinal bleeding (χ^2^ = 20.253 *p* = 0.0002) and gastric or duodenal ulcers (χ^2^ = 13.068 *p* = 0.0007). In 24.2% (n = 16) of respondents using two NOA drugs including ASA, significantly more frequent gastrointestinal bleeding and gastric or duodenal ulcers were observed (χ^2^ = 6.618 *p* = 0.01). Respondents purchasing ASAs outside the pharmacy were more likely to report allergic reactions (χ^2^ = 4.001 *p* = 0.045). ADRs occurred significantly more often in those who were unaware of the existence of the same active ingredients under different brand names. In this group, 13.4% (n = 19) were significantly more likely to experience gastrointestinal bleeding or gastric or duodenal ulcers (21.1%, n = 4) (χ^2^ = 20.096 *p* = 0.002) and gastrointestinal disorders (26.3%, n = 5) (χ^2^ = 11.025 *p* = 0.001). Gastrointestinal disorders were significantly more common among those aged 50–60 (χ^2^ = 6.500 *p* = 0.039), while respondents aged 61–75 were significantly more likely to have skin allergic reactions (χ^2^ = 11.191 *p* = 0.004). The survey showed that a small percentage of respondents used ASA in the prevention of heart disease. 

### 3.4. Analysis of the Safety of Ketoprofen Therapy

The use of ketoprofen was declared by 44.4% (n = 63) of respondents, several times a week by 12% (n = 17) and less than once a week by 32.4% (n = 46). It was used most often by people aged 50–60. Reported ADRs associated with ketoprofen therapy included gastrointestinal disorders, gastrointestinal bleeding, gastric ulcers, duodenal ulcers, allergic reactions, and renal dysfunction. In 8.5% (n = 5) of respondents taking dietary supplements and ketoprofen, gastrointestinal disorders were significantly more common (χ^2^ = 5.073 *p* = 0.024). Respondents taking two NOA drugs (9.1%, n = 6) significantly more often reported gastrointestinal disorders (χ^2^ = 5.029 *p* = 0.025). 

### 3.5. Analysis of the Safety of Diclofenac Therapy

Diclofenac was taken by only 31.7% (n = 45) of respondents, with 26.8% (n = 38) taking it less than once a week and 4.9% (n = 7) taking it several times a week. The 76–90-year-olds were the least likely to use it. ADRs were reported by 24.4% (n = 11) of respondents. The most common were gastrointestinal disorders, indicated by 45.5% (n = 5) of respondents. Figure 3 provides a summary of the ADRs of the most commonly used NOAs. 

## 4. Discussion

Aging is a gradual and irreversible pathophysiological process that manifests as a decline in tissue and cell function, and with a significant increase in the risk of various diseases associated with aging (neurodegenerative diseases, cardiovascular diseases, metabolic diseases, musculoskeletal diseases and immune diseases [20]). The increase in morbidity promotes large geriatric syndromes [21]. Multimorbidity is associated with a high number and type of multi-drug therapies in general populations, often administered by multiple specialists [22]. Furthermore, the elderly use OTC medications and dietary supplements purchased on their own [23]. Our survey found that, in Poland, older adults and the elderly misuse NOAs such as paracetamol, ibuprofen, ASA and ketoprofen. We have shown that older patients take multiple drugs simultaneously, leading to a synergistic effect. Consequently, toxicity and the risk of ADRs increases. The reasons for taking medications on their own outside of the doctor’s indications were headaches or toothaches, fevers and colds, and joint diseases. Similar results were obtained by Weiner et al. [24]. According to their observations, the most popular drug in the elderly population was also paracetamol (52.5%), followed closely by ibuprofen and ASA (36.1%), metamizole (11.5%), naproxen (6.6%), and diclofenac (4.9%). The reasons for their use were headache and migraine (67.2%), rheumatic pain (54.1%), and pain associated with fever (19.7%). Rule et al. conducted a survey of 38,928 people with a median age of 56 years, finding that the most commonly used NOAs were diclofenac (35%), ketoprofen (35%), ASA (23%), ibuprofen (21%), nimesulide (15%) and naproxen (13%). They were taken for osteoarthritis (33%), pain of unspecified origin (30%), post-traumatic pain (18%), prevention of coronary artery disease (18%), migraine (7%) and fever (4%) [25]. Thus, the data are universal, and the scale of the problem is very similar regardless of nationality.

The risks associated with the use of the aforementioned drugs include primarily adverse reactions. It has been found that 2800 people die annually in Poland due to chronic use of non-steroidal anti-inflammatory drugs. Fialova et al. showed that most ADRs among seniors in long-term care were for and non-opioid analgesics [26]. Our study found that the geriatric population showed an increased incidence of gastrointestinal bleeding and gastrointestinal disorders associated with paracetamol use, especially several times a week. ADRs occurred significantly more often in individuals who were unaware of the presence of acetylsalicylic acid in preparations under various brand names. Among respondents taking ASA regularly, gastrointestinal bleeding was more common when combined with paracetamol, and gastric or duodenal ulcers were reported when it was combined with other analgesics. Most common ADRs associated with taking ketoprofen included allergic reactions and renal impairment. The risk of gastrointestinal disorders increased in patients taking dietary supplements. Polypharmacy is observed in about 26.3–39.9% of seniors in Europe and Israel, but the phenomenon is expected to grow [27]. Increased frequency of drug interaction occurs when the patient’s overall health status is not taken into consideration when determining dosing regimens, existing diseases or concomitant use of other drugs [28]. Morphological and functional changes in the body of the elderly can cause changes in pharmacokinetics, pharmacodynamics and ultimate response to the therapy used [29]. Pharmacokinetic changes include decreased renal and hepatic clearance and increased volume of distribution of fat-soluble drugs (hence increased elimination half-life). Pharmacodynamic changes typically include increased sensitivity to several classes of drugs, such as anticoagulants, cardiovascular drugs and psychotropic drugs. It is therefore necessary to take measures to improve the efficacy and safety of geriatric treatment, especially regarding the NOAs so often used in this age group [30]. 

The reason for improper pharmacotherapy may be patients’ insufficient compliance with the drug leaflet. Older patients declare that the problem is that font used in such leaflets is too small and that these contain unclear formulations regarding the use of drugs [28]. In our study, we revealed that as many as 82% of respondents read the leaflet. Observations of 262 people in New South Wales, Australia, showed that 64.5% of respondents understood the information on the maximum daily dose, while 18.7% of them took too much and 12.5% too little. Only 2% of respondents were unaware of the possibility of complications after using OTC drugs. Some, however, were unable to recognize the discomforts associated with taking the drug [31]. Vega-Moralesab et al. found that patients were reluctant to follow the indications eliminating gastrointestinal and cardiovascular risks, which should be followed by elderly patients using prescription NOAs [32]. 

Importantly, our results revealed that 95.1% of respondents purchased drugs from a pharmacy. The data show that although patients mainly get their medications from the pharmacy, they relatively rarely feel the need to educate themselves and talk to a pharmacist about possible drug interactions and ADRs. In addition, they mainly inform physicians (75%) about ADRs, but rarely pharmacists (20%) and nurses (5%), which may indicate, on the one hand, a lack of widespread knowledge of this possibility, and on the other hand, insufficient involvement of pharmacists in patient education activities. Raynor et al. confirmed the preference of patients for direct contact and receiving information on therapies from medical personnel, rather than from drug leaflets or advertisements [33]. Unfortunately, the results of a study by Chlebowska et al. conducted in Poland are alarming [34]. It has been shown that less than half of patients visiting pharmacies declare that they receive proper information from professional staff regarding the storage and use of medicines. 

Despite the development of pharmaceutical care worldwide, the level of care in Poland in this regard is much lower. However, a study by Merks et al. found that pharmacists in Poland expressed a willingness to conduct medical reviews, which could have a significant impact on optimizing patient health outcomes [35]. Many studies confirm the influence of pharmacists on the safety of patients’ therapies by providing information on medication use. This is important because it is estimated that 1 in 10 elderly people experience ADRs leading to hospitalization or during a hospital stay, and pharmacist involvement could prevent these events [36]. It has been shown worldwide that pharmacists’ drug reviews and planned interventions at admission and discharge had a positive impact on medication orders in the elderly, resulting in effective and safe therapy [37]. Since the pharmacist is one of the most accessible healthcare professionals, this can relieve the burden on physicians and optimize the medication management process within the healthcare team [36]. They play a key role as an advisor when determining therapy in self-medication with OTC products [38].

### Limitations

The limitations of the survey used in the current work include the use of an electronic survey, which may have been a difficulty in the case of older people, both because they are less skilled in this area and have less access to mobile devices and the Internet. Hence, a smaller-than-planned number of people was included in the observation. The original assumption of using a paper-based survey had to change due to security concerns related to the COVID-19 pandemic restrictions.

## 5. Conclusions

Optimizing the use of NOAs by elderly patients is only possible with the full use of pharmacists’ highly specialized knowledge and skills. There is a need to involve them in the process of coordinating the medications taken by seniors. With the lack of an adequate number of geriatricians in the Polish health care system and the incidence of adverse drug reactions—especially NOAs—increasing with age, the need for patient education and pharmacotherapy analysis performed by pharmacists through drug review cannot be overestimated. The potential of pharmacists should be tapped into, as they are the most accessible group of health care professionals to patients. Mobile devices such as the computer and cell phone can be used to plan and conduct online education for older patients aged 50–75, while in the 76–90 age range, where the fewest users of these devices were reported, in-person education during a doctor’s visit or at the pharmacy as part of pharmaceutical care is necessary. The survey showed that despite easy access to pharmacists, few seniors take advantage of the opportunity to educate themselves about the medications they are taking. It is therefore important to emphasize the importance of the role of pharmaceutical care, as through medication reviews, the number of ADRs that occur can be reduced in the future, and the safety and effectiveness of pharmacotherapy for geriatric patients can be increased.

## Figures and Tables

**Figure 1 ijerph-20-04505-f001:**
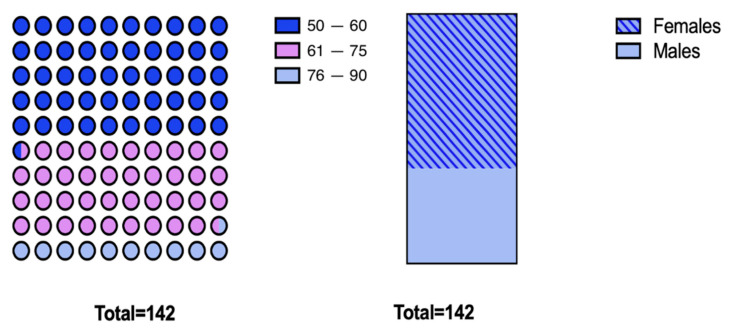
Graphic representation of the age and gender structure of the study group.

**Figure 2 ijerph-20-04505-f002:**
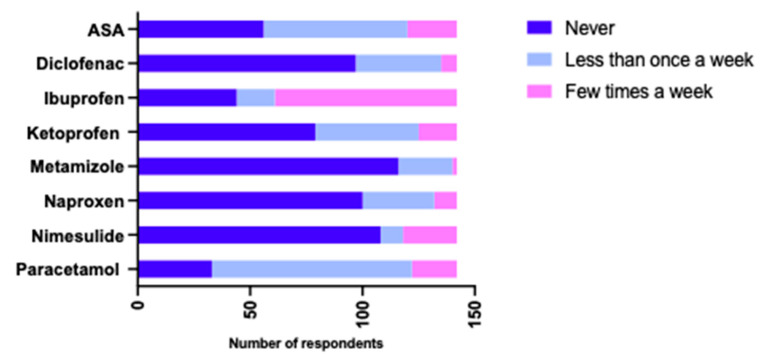
Analysis of the frequency of use of non-opioid analgesics by the elderly and older adults.

**Figure 3 ijerph-20-04505-f003:**
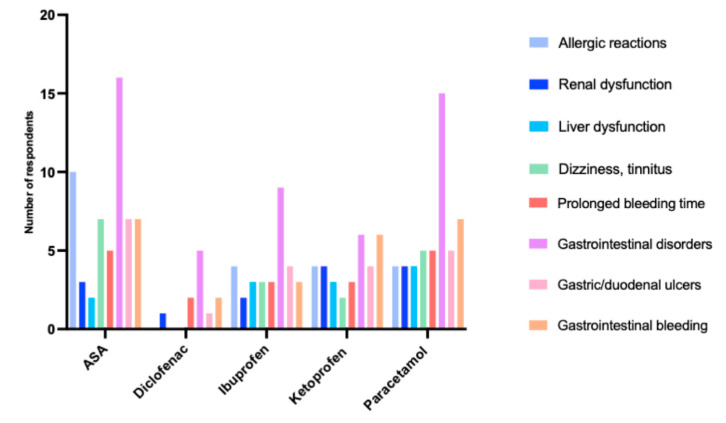
ADRs of the most commonly used NOAs in geriatric patients.

**Table 1 ijerph-20-04505-t001:** Characteristics of the study group showing the relationship of the observed diseases to the gender of the respondents.

Condition	Female Respondents (%)	Male Respondents (%)	χ2	p
Cardiovascular diseases	21.6	20.4	0.030	0.863
Hypertension	34.1	50.0	3.525	0.060
Diabetes	19.3	37.0	5.453	0.020
Kidney failure	8.0	11.0	0.401	0.527
Depression, neurosis	26.1	7.4	7.623	0.006

**Table 2 ijerph-20-04505-t002:** Characteristics of the study group showing the relationship of the observed diseases with the age of the respondents.

Condition	50–60 Years (%)	61–75 Years (%)	Over 75 Years Old (%)	χ^2^	*p*
Cardiovascular diseases	14.5	21.2	57.1	12.917	0.002
Hypertension	32.9	42.3	71.4	7.466	0.024
Diabetes	15.8	36.5	42.9	9.174	0.010
Kidney failure	6.6	5.8	35.7	13.197	0.001
Depression, neurosis	18.4	21.2	14.3	0.375	0.829
Obesity	7.9	25.0	21.4	7.318	0.026
Neoplasms	6.6	11.5	14.3	1.405	0.495

## Data Availability

Not applicable.

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
