# Peer review of "Over the Counter Pain Medications Used by Adults: A Need for Pharmacist Intervention"

_ijerph, 2023, doi:10.3390/ijerph20054505_

Round 1

Author Response

Dear reviewer, we are very grateful for the kind review. Thank You for Your interesting observations. Here the original questions and answers to Your suggestions.

Major 

Ln 2 title needs to reflect the research paper if choosing to publish as a research project. Title should have the correct sample used. For example, over the counter pain medications used by adults: A need for pharmacist intervention. So the topic of research is first, which created a need that turns into a potential solution of pharmacist engagement. 

Thank you for your comment. We agree that this title is more appropriate and we are grateful for your suggestion. We have changed the title to the suggested one.

LN 14 38 make changes that will reflect changes in text for revised paper. Introduction minimize to just the aspect of OTC pain meds and ADRs in adults/older adults. Less on history, definitions, outcomes. Keep focused on research purpose. Some info on Poland is needed to help readers understand population and pharmacy practice. Figure one is probably not needed since this paper didn t access the steps pharmacists did in their community pharmacies. This is a pharmacoepidemiology study not a health systems study. 

Thank you for your comment. We have removed the history and definitions section and rephrased the introduction. We outlined more the purpose of the work and focused on the aspect of ADRs and patients' knowledge of pharmaceutical care. 

“Due to the widespread availability of these medications and the risks associated with their incorrect use, as well as the difficulty of contacting a physician, it is necessary to effectively implement and develop pharmaceutical care, especially for elderly patients. The purpose of this study was to analyze the safety of non-opioid analgesics used by the elderly and the importance of the pharmacist in optimizing pain management in geriatric patients. In this paper, we will examine the most common adverse drug reactions associated with the use of NOAs, as well as identify factors that intensify drug toxicity in elderly patients.  We will highlight the level of knowledge of older patients regarding pharmaceutical care and their willingness to access it.” 

Ln 91 97 give more specifics about survey in methods, for example - number of items, online nature, how distributed. Some of this information is later on but needs to be here. Don t need survey title. Might consider having survey as an appendix. 

Thank you for your comment. We have added the information and attached a blank copy of the survey in appendix. 

“​​In the survey, we used a form consisting of 88 single- and multiple-choice questions to collect data from patients. The questionnaire included questions about the reasons for the use of pain medications, comorbidities, the use of dietary supplements, the source of purchasing and obtaining information about medications, and knowledge of reporting adverse drug reactions. Initially, we collected data using an online questionnaire, but due to the low online activity of people over 65, data was also collected at pharmacies.”

and

 “For the study, participants with preserved cognitive ability taking medication on their own were eligible.”

Ln 99 102 to make inferences about older adults, you will need to create a sample of > 60 or 65 years old. If you want to keep the 50 64 (50 60) people, can use that group to compare to your older adult sample. Need percent of sample that is > 60/65 years old for an older adult group. You can still give the older old percent from the older adults, but since the older old will be a smaller group might not be able to evaluate data by older adult age groups. Would split the 61 75 into < 65 and > 65 or < 60 and > 65 years. 

Thank you for your comment. As suggested, we changed the title as the analysis is about adults. Separating out respondents aged 50-60 is not necessary.

Ln 106 111 - disease conditions by sex and age are results not methods. 

Thank you for your comment. We agree that this is an unusual procedure. However,  in materials and methods, we aimed to already show that the elderly patient population is burdened with multiple chronic diseases. This has a significant impact on the increased risk of drug interactions. We therefore highlighted that the study population is at higher risk of ADRs, which we presented in the results. 

Discussion shorten. First paragraph a summary of your major findings. Next paragraphs are a comparison of your findings to other literature. Decrease amount of data in this part/ summarize. Next the need for pharmacist intervention/pharmaceutical care. 

Thank you for your comment. We have rephrased the discussion and removed unnecessary information from it. We hope that by following the comments it is now better constructed. 

Minor 

Ln 27 administering might be better selecting the 

Thank you for your comment. We've corrected it. 

Ln 36 - 38 not a conclusion of this study since value of pharmacist intervention is not accessed no outcome data from pharmaceutical care to state changed outcomes. 

Thank you for your comment. Our research has shown that in Poland very few patients know about the presence of pharmaceutical care. That's why we are highlighting it's role and directing this survey for pharmacists to come out on their own and offer help to elderly patients. 

“We are targeting pharmacists with this survey to draw attention to the problem of the prevalence of selling NOAs to geriatric patients. Pharmacists should educate seniors about the possibility of ADRs and approach patients with polypragmasy and polypharmacy with caution. Pharmaceutical care is an essential aspect in the treatment of geriatric patients, which can contribute to better results of their existing treatment and increase the safety of medication intake. Therefore, it is important to improve the development of pharmaceutical care in Poland in order to enhance patient outcomes.”

Ln 39 might want to add over-the-counter medications as an additional term

Thank you for your comment. We have added over-the-counter medications to the keywords.

Ln 46 info on organ adaptability seems out of place in this paper Poland study, what country for the studies to know how to compare 

Thank you for your comment. We have removed this sentence. 

Ln 65 69 not sure all this history is needed.

Thank you for your comment. We have removed the part about the history from the introduction. 

Ln 91 validated survey by who?

The questionnaire was not validated because our research is cognitive and informative. Validation would be necessary to obtain reliable results, e.g. for diagnosis, and to develop a questionnaire that could be used by other researchers to compare the results obtained. We have removed this information from the text. 

 Ln 99 102 methods describe who could do the survey. The description of who did the survey is the first part of results. Please move this information to the results section.

Thank you for your comment. We agree that this is an unusual procedure. However, in the materials and methods, we aimed to schacterize the population. Instead, the results regarding the head objective of the study - the effect of otc on safety - are shown in the results section. 

Ln 103 data are presented in text or table/figure; not both. The text explains the information well, so no need to include the figure. 

Thank you for the comment and we agree with your suggestion. However, we would like to leave the images in the work, because for the potential reader they allow to find information quickly and make the work more enjoyable to read. 

 Ln 136 put meds in some type of order; probably based on usage or alphabet, label x axis 

Thank you for your comment. We've corrected it. 

Ln 138 need to change figure title since you have middle and older adults, not a geriatric sample.

Thank you for your comment. We've corrected it. 

 Ln 139 75% (n=30) listed twice 

Thank you for your comment. We've corrected it. 

Ln 146 what is your definition of dietary supplements, the ones you list might be seen more as herbal products vs. dietary supplements. Add this definition to your methods. Having your survey as an appendix can also help to see how this data was captured.

Thank you for your comment. Herbal products can be classified as dietary supplements: 

https://www.degruyter.com/document/doi/10.7556/jaoa.2007.107.1.13/html 

In Poland, the same restrictions exist for herbal products and dietary supplements. 

 The authority responsible for the inspection and approval of the product for use by consumers is the sanitary inspectorate, which confirms the safety of taking supplements and herbal products.

 Ln 152 why more bleeding with expired ASA and ibuprofen, just curious vs. a need to rewrite

The reason is enhanced toxicity which is causes by products of degraded ASA and Ibuprofen.

https://www.sciencedirect.com/science/article/pii/S0731708502004004?via%3Dihub 

https://www.ncbi.nlm.nih.gov/pmc/articles/PMC4706284/ 

Ln 166 and should be in discussion, not results 

Thank you for your comment. We have removed the sentence and replaced it. 

“A self-reported survey shows that many patients did not receive proper medical interview and advice.”

Ln 225 label y axis  

Thank you for your comment. We've corrected it. 

Ln 227 change to adults vs. geriatrics

Thank you for your comment. We've corrected it. 

 Ln 252 Ln 256 since your sample is adults, need to put the ages of the older adults study groups you are comparing groups. What country do the data come from, more important than author names?  

Thank you for your comment. The discussion has been re-constructed and in the 1st paragraph there is information that the study was done in Poland

Ln 259 - What country do the data come from? more important than author names.

Thank you for your comment. The discussion has been re-constructed and in the 1st paragraph there is information that the study was done in Poland

 Ln 264 two studies do not create a universal conclusion for around the world. 

Thank you for your comment. We have removed the information that the data is universal throughout the world. 

Ln 269 did they give statistics per medications? If so only include the results for NOAs, the focus of this paper. 

Thank you for your comment and we have corrected it. 

Ln 274 was this information in the results? No new results in discussion 

Thank you for your comment. We have removed the information. 

We are very grateful for the reviews and such insightful comments. We hope that now our manuscript has improved significantly and can be published. We would like to thank you very much for your thoroughness.

Reviewer 2 Report

Thank you for the opportunity to review this manuscript titled “Polypharmacy and adverse drug reactions in geriatric patients: 2 the pharmacist's role in safe pharmacotherapy”. This is a very important area of research which is well covered by the authors. Below are the comments:

·        The most abused OTC…”. Why would the authors assume that all use is “abuse”. Are they certain that these medicines are not used as indicated?

·        The purpose of the study in unclear (Pg 3, Line 86). What do the authors mean when they say they want to analyse the use of non-opioid analgesics?

·        The introduction itself is quite generalised. Authors have focused on the demographics of the ageing population and the history of pharmaceutical care but have failed to address what their priority is for this research paper. It is recommended that authors highlight what the problem is when using analgesics, with a neutral discussion on the benefits and harms of these medicines in this population. Once established they should then highlight how pharmacist can contribute towards reducing harm. As is, it is very unclear what the authors are proposing in this study.  

·        Could the authors include the questionnaire as a supplementary material? This would provide the reader a context for the type of data collected. Currently this is vague.

·        It is unusual that authors have chosen to present part of the results as methods.

·        The first paragraph of results is quite unclear. I would suggest edits to grammar and language to highlight what the authors are trying to out across, e.g, it is unclear what they mean by “In joint disease, statistically significant women were more likely to take them than men….”.

·        “It is worth noting that 147 combining dietary supplements with NOA significantly increased the incidence of 148 ADRs. When taken with ASA and paracetamol, they manifested as gastrointestinal 149 bleeding (χ2=4.738 p=0.029), (χ2=11.672 p=0.001), while combination with ketoprofen 150 caused gastrointestinal distress (χ2=5.073 p=0.024).” What comparative analysis did the authors do to come to this conclusion?

·        It is strongly recommended that data be presented in a uniform manner, i.e., use 1 or 2 decimal places for all presentations.

·        The authors relate GI bleeding to increased use of paracetamol. Is this supported by literature?

·        “Regular smokers were more likely to 180 report gastrointestinal bleeding 7.1% (n=16), (χ2=22.846 p=0.00004), similarly alcohol 181 consumers 7.1% (n=16), (χ2=5.500 p=0.019).” How was this association established?

·        What is the purpose of Figure 4 – this information is readily available in literature. How did the authors assess renal or liver dysfunction?

·        The first 3 paragraphs of discussion has no relation to the findings of this study therefore it does not have relevance.

·        In the discussion the authors state that older people ‘abuse’ NOAs. How did they establish abuse? Did the patients respond that they are consuming this without having symptoms or achieving any symptomatic relief from these medicines?

·        “ The reasons for taking medications on their own outside of the doctor's indications were headaches or toothaches, fevers and colds, and joint diseases.” It is unclear why authors would define this as abuse.

·        Did the authors assess the cognitive ability of the participants? Also, was the medication self-consumed or the participants have a carer?

Overall the theme of the research is of importance however the survey has been poorly reported.

Author Response

Dear reviewer, we are very grateful for the kind review. Thank You for Your interesting observations. Here the original questions and answers to Your suggestions.

“The most abused OTC…”. Why would the authors assume that all use is “abuse”. Are they certain that these medicines are not used as indicated?

Thank you for your comment. We have rephrased the sentence to “Among the most commonly used OTC drugs are non-opioid analgesics, used to treat musculoskeletal disorders, fevers, inflammation or the common cold.”. We have completed the bibliography and added articles describing the prevalence of OTC drug use and indicating that NOAs are among the most commonly used drugs.

 The purpose of the study in unclear (Pg 3, Line 86). What do the authors mean when they say they want to analyse the use of non-opioid analgesics?

 Thank you for your comment. We have rephrased the sentence to “The purpose of this study was to analyze the safety of non-opioid analgesics used by the elderly and the importance of the pharmacist in optimizing pain management in geriatric patients.”.

 The introduction itself is quite generalised. Authors have focused on the demographics of the ageing population and the history of pharmaceutical care but have failed to address what their priority is for this research paper. It is recommended that authors highlight what the problem is when using analgesics, with a neutral discussion on the benefits and harms of these medicines in this population. Once established they should then highlight how pharmacist can contribute towards reducing harm. As is, it is very unclear what the authors are proposing in this study.  

Thank you for your comment. We have removed information about the history of pharmaceutical care and definitions from the introduction. We have rewritten the introduction and focused more on the purpose of the study. We have also made changes to the discussion and hope that the correct order is now followed, making the work clearer. 

  Could the authors include the questionnaire as a supplementary material? This would provide the reader a context for the type of data collected. Currently this is vague.

Thank you for your comment. We have added a blank copy of the questionnaire in aprrendix. 

  It is unusual that authors have chosen to present part of the results as methods.

Thank you for your comment. We agree that this is an unusual procedure. In materials and methods, we aimed to already show that the elderly patient population is burdened with multiple chronic diseases. This has a significant impact on the increased risk of drug interactions. We therefore highlighted that the study population is at higher risk of ADRs, which we presented in the results. 

The first paragraph of results is quite unclear. I would suggest edits to grammar and language to highlight what the authors are trying to out across, e.g, it is unclear what they mean by “In joint disease, statistically significant women were more likely to take them than men….”.

Thank You for your comment, we have rephrased the sentence to “In the population suffering from joint disease, NOAs were more frequently used by women than men.”

“It is worth noting that 147 combining dietary supplements with NOA significantly increased the incidence of 148 ADRs. When taken with ASA and paracetamol, they manifested as gastrointestinal 149 bleeding (χ2=4.738 p=0.029), (χ2=11.672 p=0.001), while combination with ketoprofen 150 caused gastrointestinal distress (χ2=5.073 p=0.024).” What comparative analysis did the authors do to come to this conclusion?

Thank You for your comment, we have rephrased the sentence. The probability of correlation between variables was supported using count tables and multivariate tables. Statistical significance was assessed using Pearson's χ2 test of concordance, taking α = 0.05 as the level of significance. In the case of expected counts less than 5, the chi-square test with Yates correction was applied additionally. 

It is strongly recommended that data be presented in a uniform manner, i.e., use 1 or 2 decimal places for all presentations.

Thank You for your comment. The presentation of the results has been changed and now 1 decimal place is used in all data.

 The authors relate GI bleeding to increased use of paracetamol. Is this supported by literature?

Thank you for your comment. In order to confirm the link between the use of medicines and specific ADRs, we have analyzed the adverse reactions included in the summary of product characteristics of medicines used in Poland. The information is provided by drug manufacturers. 

https://rejestrymedyczne.ezdrowie.gov.pl/api/rpl/medicinal-products/10723/characteristic   

 “Regular smokers were more likely to 180 report gastrointestinal bleeding 7.1% (n=16), (χ2=22.846 p=0.00004), similarly alcohol 181 consumers 7.1% (n=16), (χ2=5.500 p=0.019).” How was this association established?

Thank you for your comment. We performed tests for the non-smoking and non-alcohol consuming population and compared the results obtained with the population of smokers and those consuming alcohol products.  

What is the purpose of Figure 4 – this information is readily available in literature. How did the authors assess renal or liver dysfunction?

The purpose of Figure 4 was to illustrate the data included in the paper for an easier search for specific ADRs of particular drugs. We wanted to make it easy for the reader to infer correlations despite a large amount of data, which would increase the clarity of the presented results. 

The first 3 paragraphs of discussion has no relation to the findings of this study therefore it does not have relevance.

Thank you for your comment. The purpose of the first three paragraphs was to show why we chose this study group. We wanted to outline that the problem of polypragnosia occurs mainly in the elderly. In addition, the elderly are more vulnerable to drug toxicity. This is why it is important for pharmacists to specifically monitor self-medication in the elderly population.  

 In the discussion the authors state that older people ‘abuse’ NOAs. How did they establish abuse? Did the patients respond that they are consuming this without having symptoms or achieving any symptomatic relief from these medicines?

Thank you for your comment. In the survey, we asked about the use of specific NOAs. In Poland, these drugs are available without a prescription, and most patients take them on their own initiative without consulting a doctor (almost 90%). 

https://advances.umw.edu.pl/en/article/2016/25/2/349/ 

Based on the survey, we inferred that patients are combining the drugs which enhances their effect. Furthermore, older patients are unaware that a substance exists under different brand names and will often take the same drug several times. 

  “ The reasons for taking medications on their own outside of the doctor's indications were headaches or toothaches, fevers and colds, and joint diseases.” It is unclear why authors would define this as abuse.

Thank you for your comment. We defined this as abuse because patients were taking several drugs simultaneously without a doctor's indication. As a result, they were increasing the potency of the drugs through synergism of their effects, leading to ADRs. We clarified this in the discussion. 

“We have shown that older patients take multiple drugs simultaneously, leading to a synergistic effect. Consequently, toxicity and the risk of ADRs increases.“

  •       Did the authors assess the cognitive ability of the participants? Also, was the medication self-consumed or the participants have a carer?

Thank you for your question. We have added this information in the paper "For the study, participants with preserved cognitive ability taking medication on their own were eligible. “

Overall the theme of the research is of importance however the survey has been poorly reported.

We are very grateful for the reviews and such insightful comments. We hope that now our manuscript has improved significantly and can be published. We would like to thank you very much for your thoroughness.
